# Quantitative Risk Assessment of *Bacillus cereus* Growth during the Warming of Thawed Pasteurized Human Banked Milk Using a Predictive Mathematical Model

**DOI:** 10.3390/foods11071037

**Published:** 2022-04-02

**Authors:** Miroslava Jandová, Pavel Měřička, Michaela Fišerová, Aleš Landfeld, Pavla Paterová, Lenka Hobzová, Eva Jarkovská, Marian Kacerovský, Milan Houška

**Affiliations:** 1Tissue Bank, University Hospital Hradec Králové, 500 05 Hradec Králové, Czech Republic; pavel.mericka@fnhk.cz (P.M.); michaela.fiserova@fnhk.cz (M.F.); 2Department of Histology and Embryology, Faculty of Medicine in Hradec Králové, Charles University, 500 03 Hradec Králové, Czech Republic; 3Food Research Institute Prague, 102 00 Prague, Czech Republic; ales.landfeld@vupp.cz (A.L.); milan.houska@vupp.cz (M.H.); 4Department of Clinical Microbiology, University Hospital Hradec Králové, 500 05 and Faculty of Medicine in Hradec Králové, Charles University, 500 03 Hradec Králové, Czech Republic; pavla.paterova@fnhk.cz; 5Department of Hospital Hygiene, University Hospital Hradec Králové, 500 05 Hradec Králové, Czech Republic; lenka.hobzova@fnhk.cz; 6Department of Pediatrics, University Hospital Hradec Králové, 500 05 Hradec Králové, Czech Republic; eva.jarkovska@fnhk.cz; 7Department of Obstetrics and Gynecology, University Hospital Hradec Králové, 500 05 Hradec Králové, Czech Republic; marian.kacerovsky@fnhk.cz

**Keywords:** *Bacillus cereus*, human pasteurized milk, predictive microbiology, mathematical growth model

## Abstract

*Bacillus cereus* is relatively resistant to pasteurization. We assessed the risk of *B. cereus* growth during warming and subsequent storage of pasteurized banked milk (PBM) in the warmed state using a predictive mathematical model. Holder pasteurization followed by storage below −18 °C was used. Temperature maps, water activity values, and *B. cereus* growth in artificially inoculated PBM were obtained during a simulation of manipulation of PBM after its release from a Human Milk Bank. As a real risk level, we chose a *B. cereus* concentration of 100 CFU/mL; the risk was assessed for three cases: 1. For an immediate post-pasteurization *B. cereus* concentration below 1 CFU/mL (level of detection); 2. For a *B. cereus* concentration of 10 CFU/mL, which is allowed in some countries; 3. For a *B. cereus* concentration of 50 CFU/mL, which is approved for milk formulas. In the first and second cases, no risk was detected after 1 h of storage in the warmed state, while after 2 h of storage, *B. cereus* concentrations of 10^2^ CFU/mL were occasionally encountered. In the third case, exceeding the *B. cereus* concentration of 10^2^ CFU/mL could be regularly expected after 2 h of storage. Based on these results, we recommend that post-pasteurization bacteriological analysis be performed as recommended by the European Milk Bank Association (EMBA) and using warmed PBM within 1 h after warming (no exceptions).

## 1. Introduction

Over the past years, advances in neonatal intensive care have improved the survival of very-low-birth-weight neonates [1]. Feeding with pasteurized banked milk (PBM) has been regarded as an essential part of neonatal intensive care as it supports neurocognitive development [2,3], reduces the risk of obesity in later life [4], and protects cardiovascular health in adulthood [5]. Lowering the incidence of necrotizing enterocolitis in newborns fed PBM compared to that in babies fed milk formula is a significant advantage for this type of feeding [6]. On the other hand, due to the immaturity of the immune system in extremely-low-birth-weight neonates, they are at risk of opportunistic infections, such as *B. cereus* [7,8,9,10,11,12]. While in healthy newborns, feeding with PBM is temporary support in cases of the mother’s delayed or insufficient lactation, in very-low-birth-weight neonates, PBM represents the only option because breastfeeding is not possible. Preferably the PBM used is expressed by their mothers, with donated PBM being the second option. Standard Holder pasteurization followed by rapid chilling and/or freezing of human milk is recommended by the EMBA, which also recommends routine post-pasteurization bacteriological analysis [13]. Such analysis is not yet obligatory in the Czech Republic [14], nevertheless, Czech Human Milk Banks are obliged to perform regular evaluations of their pasteurization process as a part of the Hazard Analysis and Critical Points system (HACCP) and to perform corrective actions, as necessary [15]. Many human milk banks perform pre-pasteurization pooling of milk and take a post-pasteurization sample representing a large volume of the pooled milk [7,16]. The quality assurance system of our Human Milk Bank is based on avoidance of pooling and taking samples from each pasteurized bottle with a volume of 250 mL [15].

To facilitate risk evaluation, the predictive microbiology databases COMBASE and Pathogen Modeling Program are available. Lewin et al., 2020 [16] used a different prediction approach based on the Monte Carlo method using statistical data from disease cases and determined the probability of infection per 1 million doses of PBM. Lewin’s simulation study [16] remained limited to the safety of PBM at the moment of its release for distribution. An evaluation of *B. cereus* growth curves in milk products under different conditions based on experimental data and the COMBASE Predictive Models was published by Soleimaninanadegani [17]. Results of mathematical modeling of *B. cereus* growth in milk were also published by Teleken et al. [18], Ačai et al. [19], Larsen et al. [20], and Hyoun Wook, K. et al. [21]. They published predictive models to evaluate the kinetic behaviors of *Bacillus cereus* and *Staphylococcus aureus* in milk during storage at various temperatures.

In our previous retrospective study based on data from 2017–2020, we identified *B. cereus* as the most common cause of post-pasteurization PBM discard in our Human Milk Bank [15]. As extension of the previous investigation, we deal with assessing the residual risk that may occur during the warming of the thawed PBM and during temporary storage in the warmed state prior to administration to newborns at neonatal intensive care units (NICU).

In this paper, we used a predictive mathematical model created on real temperature history data, water activity, and *B. cereus* growth in artificially inoculated PBM to assess this risk. The risk was assessed for three cases: 1. An initial post-pasteurization *B. cereus* concentration of 1 colony-forming unit (CFU)/mL, which is equal to the level of detection; 2. a *B. cereus* concentration of 10 CFU/mL, which is allowed in some countries [13]; and 3. a *B.cereus* concentration of 50 CFU/mL, which is approved for milk formulas [22,23]. The actual risk level was set at 100 CFU of *B. cereus*/mL.

As some authors state, precise quantification of risks is more important than in the context of food manufacturing alone [24]. The originality of our work is in quantitative risk assessment of *B. cereus* growth in thawed human pasteurized milk during manipulation at neonatal units of Pediatric departments.

## 2. Materials and Methods

### 2.1. The Description of Practice before and after Delivery of PBM to Pediatric Departments

#### 2.1.1. Human Milk Collection and Pasteurization

After collection and input analysis at the Human Milk Bank, standard Holder pasteurization at 62.5 °C for 30 min is performed. The warming phase is followed by rapid chilling in the cooling section of the pasteurization device. The milk is divided into sterile 100 mL distribution bottles in the laminar flow cabinet, and milk samples for post-pasteurization microbiological analysis are taken. Then, the milk is rapidly frozen in a blast freezer and stored in quarantine at −20 °C until it is released for clinical use by a qualified person at the Human Milk Bank. Technology details are presented in our previous paper [15].

#### 2.1.2. Release of PBM for Clinical Application, Storage of Released PBM, and Instruction for Its Use

Only PBM with negative post-pasteurization microbiological findings is released for clinical use. The released PBM is stored in a freezer at −27 °C (Liebherr Premium, Liebherr Hausgeräte, Ochsenhausen, Germany) for no longer than three months. Released PBM is distributed upon demand to clinical departments daily, except on weekends; PBM is intended to be used within one hour after thawing and warming to body temperature.

#### 2.1.3. Delivery of PBM from the Human Milk Bank and Manipulation after Delivery

Human Milk Bank nurses deliver PBM. Transport to the Department of Pediatrics or Obstetrics and Gynecology is performed in insulated plastic boxes used exclusively for this purpose. In clinical wards, PBM is stored in a freezer (Zanussi, Vallenoncello, Italy) in until thawing. Thawing is performed either slowly in the refrigerator (Tefcold, Viborg, Denmark) or rapidly in a warming device (Beurer, Baby care, Ulm, Germany). PBM is warmed in a water bath set at 45 °C until it reaches body temperature, usually in 20–30 min. According to Human Milk Bank instructions, the warmed milk must be used within one hour.

#### 2.1.4. Feeding PBM to Newborns

Feeding warmed PBM to newborns is usually performed at 3 h intervals by clinical department nurses. In the NICU with preterm newborns, the warmed PBM is transferred into 10 mL sterile disposable syringes and administered via a gastric catheter. Usually, 5–8 premature newborns are fed from one 100 mL distribution bottle. Feeding newborns using baby bottles or syringes is performed in newborns of normal weight. The amount of one feeding is 30–40 mL.

A schematic description of the transport and administration of PBM is presented in Figure 1.

### 2.2. Simulation of PBM Manipulation after Delivery from the Milk Bank

The design of the experiment is presented in Figure 2.

#### 2.2.1. Temperature History I Slow Thawing of Frozen PBM in the Refrigerator

In the first phase, temperature measurements of slow thawing were performed simultaneously in 4 bottles of 100 mL frozen PBM, which were placed in the lower part of the refrigerator (Liebherr Premium, Liebherr Hausgeräte, Ochsenhausen, Germany) with a temperature range of 2–8 °C. A thermocouple was placed in the middle of each bottle. The air temperature in the refrigerator was measured with a COMET S3120 (Comet System Ltd., Rožnov pod Radhoštěm, Czech Republic) memory thermometer, and the temperature profiles inside the bottles were measured with a 4-channel memory thermometer COMET M1200E (Comet System Ltd., Rožnov pod Radhoštěm, Czech Republic). The layout of the bottles within the refrigerator is shown in Figure 3. The measurement lasted 20 h.

#### 2.2.2. Temperature History II Warming of Bottles Containing Thawed PBM

In the next phase, the temperatures inside the bottles containing the thawed milk were measured during warming in the water bath, routinely used for warming thawed milk before administration to newborns. Measurements were performed in two separate bottles from the previous experiment (bottles No. 2 and 4). A water bath heater (Canpol babies, Canpol Sp., zo. o, Warsaw, Poland) was used for warming (Figure 4). Figure 4 also shows the location of the temperature sensors (thermocouples). The water bath temperature, the temperature on the outer surface of the bottle, the temperature of the milk at the inner surface of the bottle, and the temperature of the milk in the middle of the bottle were measured simultaneously.

#### 2.2.3. Water Activity and pH Measurement

Water activity was measured in 3 samples of thawed PBM using an Aw SPRINT TH-500 (Novasina AG, Lachen, Switzerland). The same samples were also used for pH measurement (Testo, Lenzkirch, Germany).

#### 2.2.4. Experiment with PBM Inoculated with *B. cereus*

A total of 3 bottles of PBM were inoculated with *B. cereus*. The bacilli strain was isolated from PBM during routine testing in 2021. The *B. cereus* strain used in this study was randomly selected from the *B. cereus* strains isolated from bottles of PBM revealing post-pasteurization *B. cereus* positivity determined by MALDI-TOF MS (Bruker Daltonics, Hamburg, Germany). The common feature of all isolates was that they were well-adapted to all processing steps of the Human Milk Bank, i.e., initial storage in a refrigerator, standard Holder pasteurization followed by rapid chilling, shock freezing, and storage at −27 °C. The selected strain was stored in an ultra-freezing box at −70 °C ± 5 °C. After revitalization, the strain was suspended in sterile water, and 1 mL of suspension was inoculated into each milk bottle. The final inoculum in the milk bottle was calculated to be 1–5 × 10^3^ CFU/mL, and the final inoculum was verified using the European Pharmacopoeia (EPh) 2.6.12 [25]. The bottles inoculated with *B. cereus* were placed into a water bath (Canpol babies, Canpol Sp., zo. o, Warsaw, Poland) adjusted to +45 °C. The milk samples for bacteriological testing were collected after 15, 60, 120, and 180 min. Two samples were taken from each bottle, one from the bottom of the bottle and the other from the upper layer of milk just below the surface. Bacterial counts were determined according to EPh 2.6.12 (see above).

#### 2.2.5. Quantitative Assessment of *B. cereus* in PBM

For this study, assessments were carried out in accordance with the European Pharmacopoeia (EPh) Chapter 2.6.12 [25]. A total of 500 μL of PBM was inoculated directly on two plates of Columbia agar (Oxoid Ltd., Basingstoke, UK), and the agars were incubated under ambient atmosphere and at 35 ± 2 °C. After 18–24 and 48 h, CFUs (colony-forming unit) per milliliter for each milk sample and time were determined.

#### 2.2.6. Mathematical Modeling of *B. cereus* Growth

Temperature history data during thawing and warming of PBM are presented as Appendix A. American Predictive Software, Pathogen Modeling Program version 8 (Agriculture Research Service, Washington, DC, USA) [26] was used for mathematical modeling of microorganism growth. This software allows to predict the growth of microorganisms during different conditions. The models are based on extensive experimental data of microbial behavior in liquid microbiological media and food. The growth model *B. cereus*—vegetative forms in Broth Culture was used for our predictions. The necessary data for growth include temperature history, water activity, and pH values. The results of the prediction are then the growth curves for the given conditions. Based on these predicted growth curves, appropriate conclusions were drawn regarding the shelf life of human milk after warming of thawed PBM to the delivery temperature. Authors of the model used three strains of *B. cereus* for its construction: an emetic toxin-producing strain isolated by R. Gilbert from cooked rice; B4AC, a diarrheal toxin-producing strain isolated by D. Mossel from pea soup; and T, a reference strain (all furnished by F. Busta) [27]. Predictions for the maximum storage time of 3 h at +42 °C (maximal optional temperature of this model) were made for pH and water activity of inoculated milk. Values of pH and water activity are presented in Table 1. The software predicted a minimum initial concentration of *B. cereus* of log 3 CFU/mL. This predicted dependence calculated percent increases for 1, 2, and 3 h. However, we needed an initial concentration of 1 to 100 CFU/g for our predictions. Therefore, based on the assumption that the increase in 3 h will be within the exponential growth phase, a shift of the growth curve was made to these lower initial concentrations. This shift can be made only if the microbial concentration is low and far from almost constant (no growth part of the curve). The predictions were made for initial concentrations, which were determined by microbial examination in experiments with PBM inoculated with *B. cereus* at initial warming times (Appendix A). Other predictions were made for (1) the initial post-pasteurization *B. cereus* concentration of 1 CFU/mL, which was equal to the level of detection for *B. cereus,* (2) a concentration of 10 CFU/mL, which is a limit allowed in some countries, and (3) for a *B. cereus* concentration of 50 CFU/g, which is the approved limit for powdered milk formulas. Water activity 0.99 and pH = 6.6 were used for these PBM predictions.

#### 2.2.7. Quantitative Assessment of *B. cereus* Growth during PBM Warming

A quantitative microbial examination was performed in three runs of rapidly warming the bottles with PBM thawed slowly in the refrigerator for 9 h. The air temperature in the refrigerator was measured by data loggers (TESTO) placed on the same shelf as the bottles. The bottles were put into the heater and stored there for 3 h. Sampling for quantitative bacteriology tests was performed in a laminar flow cabinet immediately after slow thawing and during the warming process after 15 min and 1, 2, and 3 h. For quantitative bacteriology testing, milk samples (2 mL) were put into sterile test tubes.

## 3. Results

### 3.1. Slow Thawing Temperature History of Frozen PBM in the Refrigerator

The temperature history of slow thawing is presented in Figure 5. Bulleted lists look like this:

### 3.2. Warming Temperature History of Bottles Containing Thawed PBM

Time–temperature history during the warming of thawed PBM (bottle No. 2 and No. 4) is described in chapter No. 2.2.2 and presented in Figure 6.

### 3.3. Water Activity and pH Measurement

Water activity and pH values are presented in Table 1.

**Table 1 foods-11-01037-t001:** Water activity and pH of thawed PBM samples.

PBM Sample No.	Water Activity	pH
1	0.99	6.61
2	0.99	7.29
3	0.99	6.62

### 3.4. Results of the Experiment with Thawed PBM Inoculated with B. cereus

A summary of the results is shown in Figure 7. The results show that in sample No. 2 (green line), *B. cereus* growth was significant. After 60 min, an increase of one order of magnitude was detected in the sample taken from the bottom, and after 120 min, it had increased by two orders of magnitude.

### 3.5. Quantitative Assessment of B. cereus CFU during Warming

Changes in the *B. cereus* CFU/mL count during warming are presented in Table 2.

The milk in bottle MM7595A presented no growth, while at sample No. MM7599A, there was growth up to 25 CFU/mL after 180 min.

### 3.6. Prediction of B. cereus Growth

Based on the data presented above, the growth of *B. cereus* in pasteurized human milk was modeled. As initial *B. cereus* concentrations (CFU/mL), we used the data assessed in our previous study, where 80% of the samples had a quantity lower than 10 CFU/mL [15]. The resulting *B. cereus* growth models are included in the Appendix A. The results showed the calculated CFU/mL values and the upper and lower uncertainty limits.

### 3.7. Risk Assessment Using the Predictive Mathematical Model for the Case of Keeping Existing Standard Operational Protocol (SOP) with the Total Post-Pasteurization Bacteriological Analysis

The total number of 100 and 50 mL PBM bottles that were processed and analyzed according to existing SOPs delivered to Pediatric Department Units from the Human Milk Bank in the period 2017–2020 is presented in Table 3 and Table 4. The Tables show that approximately one third of delivered 100 mL PBM bottles and three fourths of delivered 50 mL bottles were used at neonatal intensive care units (NICU).

The results of predictions for the initial post-pasteurization *B. cereus* concentration of 1 CFU/mL, which is equal to the limit of detection, are presented in Table 5 and Table 6.

Table 5 shows the prediction of *B. cereus* CFU concentrations after 1, 2, and 3 h of storage at +42 °C. The risk concentrations occasionally occurred after 2 and 3 h of warming. Table 6 shows the predictions for 2 and 3 h of storage at +42 °C. Table 6 presents the total *B. cereus* CFU number ingested per day by premature newborns in cases where 10 mL of PBM was thawed and warmed per one feeding with a feeding frequency of every 3 h intervals. Table 6 assumes an initial post-pasteurization concentration of 1 CFU/mL before warming of thawed PBM.

The data in Table 6 show that the administration of warmed PBM can be regarded as safe only if storage in the warmed state does not exceed one hour. After 2 h of storage at +42 °C, *B. cereus* numbers higher than 1000 CFU per day can occasionally be ingested. After 3 h of storage at +42 °C, the limit is regularly exceeded.

### 3.8. Risk Assessment in Post-Pasteurization B. cereus Concentrations Higher than 1 CFU/mL

The data on initial *B. cereus* post-pasteurization CFU quantities available in our previous paper [15] showed that in most cases, the concentrations were low, in 80% of cases less than 10 CFU/mL and in 90% of cases less than 25 CFU/mL. However, occasionally, concentrations as high as 100 *B. cereus* CFU/mL were found. Table 7 shows the prediction results for initial post-pasteurization concentrations of *B. cereus* of (1) 10 CFU/mL (as it is allowed in some countries), (2) 50 CFU/mL, which is a limit for milk formulas, and (3) the highest concentration found in our study, i.e., 100 CFU/mL after storage at +42 °C for 1, 2, and 3 h.

After 1 h of storage in the warmed state, dangerous concentrations of *B. cereus* of 10^2^ CFU/mL were not regularly predicted for initial CFU concentrations of 10 and 50 CFU/mL. When the initial concentration of *B. cereus* was 50 CFU/mL, risk occasionally occurred (Table 7). However, after 2 h of storage, dangerous concentrations were achieved in both initial concentrations of 10 and 50 CFU/mL. After 3 h of storage, at-risk concentrations of *B. cereus* of 100 CFU/mL were regularly exceeded (Table 7).

Table 8 presents the total *B. cereus* CFU number ingested per day by premature newborns when they consumed 10 mL of thawed and warmed PBM per one feeding, and the feeding frequency was at 3 h intervals. Before warming of PBM, the initial post-pasteurization concentration was 10 or 50 CFU/mL.

## 4. Discussion

The traditional approach to evaluating the quality and microbiological safety of food products is based on a comparison of results of the bacteriological output analysis with limits set by the manufacturer, published data of manufacturers of similar products, and assessment of the compliance rate with existing national or international norms or recommendations. Such an approach was partly applied in our previous paper in which we compared the post-pasteurization discard rate with similar data published by French Human Milk Banks (HMBs) [15,22,28]. Such a comparison can provide relevant data only when similar processing and analysis methods are used. From this point of view, our data compares well with HMBs that use standard Holder pasteurization and MALDI mass spectrometry for post-pasteurization microbiological analysis. Cormontagne [22] reported that PBM discard rates, after introducing the MALDI method, fluctuated between 19.0 and 21.2%, while our discard rate fluctuated between 8.6 and 10.5%, with *B. cereus* being responsible for 67.24% of the discard. Cormontagne [22] reported that as much as 90% of PBM discard was caused by *B. cereus*. Noncompliance rates as high as 27.3%, mainly caused by *B. cereus,* were reported by Adjidé, 2019 [29]. Lewin, 2019 reported that the discard rates between 25 and 35% were typical in Canadian Human Milk Banks, and the proportion caused by *B. cereus* was between 80% and 90% [16]. Mallardi recently published a review of complex measures that substantially lowered the PBM discard rate from 19.5% to 14.3% [30].

Another critical factor is the manufacturer’s post-pasteurization discard limit. A limit of below 1 CFU/mL was used in our previous paper [15] and was equal to the limit published by Cormontagne [22], Adjidé [29], and Lewin [16]. Using this limit, Lewin predicted that the real *B. cereus* concentration fluctuates between 0.21 and 0.64 CFU/mL.

The results of quantitative post-pasteurization analysis published previously [15] were used for comparison with different *B. cereus* contamination limits included in HMBs standards or food norms. Table 9 shows 80% compliance of these samples with existing HMB microbiological safety post-pasteurization standards and 90% compliance with the norms set for food used for children under six months (Table 9). These data document the high microbiological safety of our PBM at the moment of delivery from the HMB.

Another approach to evaluating microbiological safety uses the limits for the total number of ingested *B. cereus* CFUs reported as dangerous. A range for the *B. cereus* ingested load of between 10^5^ and 10^8^ CFUs was presented by Vidic [31], while other sources [32] regard amounts as low as 10^3^ to 10^4^ as hazardous. A similar approach based on determining the maximum number of *B. cereus* CFUs ingested per one feeding was also applied in the simulation study presented by Lewin [11]. Other authors report dangerous concentrations of *B. cereus* CFUs per 1 g of food. Bacterial counts of 10^5^ to 10^8^ CFU/g of food can generate disease-relevant amounts of toxins in foods or the small intestine [33]. Most food-borne outbreaks caused by the *B. cereus* group have been associated with bacterial concentrations above 10^5^ CFU/g of foodstuff. However, it is important to highlight that both emetic and diarrheal disease have been reported for *B. cereus* counts of 10^3^ to 10^5^ CFU/g [34].

There is a consensus that even post-pasteurization microbiological negativity of PBM does not eliminate the risk of its use. During feeding of newborns, the above-presented data of *B. cereus* concentration and/or of the total amount of CFUs ingested should not be exceeded.

The methods of predictive microbiology are generally regarded to be a standard tool for assessing the risk of food, the properties of which are changing over time because of the presence of viable bacteria. We used this method in the past for predicting the growth of several microbes, including *B. cereus,* during the processing of human milk [35] and for evaluating thermo- and baro-inactivation of *E. faecium* and *St. epidermidis* in human and cow milk [36,37]. This paper applies this method to simulate manipulations with thawed PBM in neonatal hospital wards.

The presented prediction showed different results when the maximal acceptable *B. cereus* concentration of 100 CFU/mL was used (Table 5, Table 6, Table 7 and Table 8), or if the total number of 1000 CFU/mL of *B. cereus* ingested per day was used as the acceptable upper limit (Table 6 and Table 8) (the total *B. cereus* CFU numbers ingested per day shown in Table 6 and Table 8 were calculated for a total amount of 80 mL of PBM used daily for the feeding of premature newborns). Based on the *B. cereus* CFU/g limit, the first approach showed a relatively wide range of safe initial *B. cereus* CFU post-pasteurization concentrations and warming times. When using our SOP discard criteria (Table 9) and our instructions to use PBM within 1 h of thawing [15] (Table 5), this limit was not exceeded even after 2 h of warming (Table 6); moreover, Table 7 demonstrated that even PBM with initial post-pasteurization *B. cereus* concentrations of 10, 50, and 100 CFU/mL PBM used within 1 h of warming could be considered relatively safe; however, 2 or 3 h of warming PBM with these initial *B. cereus* CFU post-pasteurization concentrations, which are compliant with the general food standards (Table 9), should not be considered safe.

When using the second approach, i.e., the total limit of *B. cereus* CFU ingested per day (Table 6 and Table 8), the range of both safe initial *B. cereus* CFU/mL concentrations and warming times were much narrower.

PBM compliant with our SOP was again proven safe if used within 1 h of warming. After 2 h, the total *B. cereus* CFU limit could occasionally be exceeded (Table 6). In case of using the PBM with an initial concentration of 10 CFU/mL, marginal total *B. cereus* CFU numbers could be expected after one hour of warming as shown in Table 8. PBM with initial concentrations of 50 CFU/mL, even if used within 1 h, produced unacceptable total CFU counts (Table 8).

Our calculations probably overestimated the risk for three reasons. The first is that the risk was calculated for a 100% proportion of *B. cereus* in the total PBM discard, while in reality, this proportion can vary between 70% [15] and 90% [22]. The second reason is our predictive mathematical model’s wide uncertainty limit range (Table 5, Table 6, Table 7 and Table 8). The third reason may be a prolonged lag phase in microbes damaged by the processing techniques described in our previous paper, i.e., pasteurization followed by rapid chilling and freezing [15]. This is supported by *B. cereus* growth diagrams (Appendix A). Nevertheless, the results of the presented predictions (Table 5 and Table 6) support the current practice of HMBs and their post-pasteurization discard limits [13] (Table 9) and confirm the high level of safety of our past and current practices [15]. In this study oriented to the needs of the routine practice of HMBs, we did not consider the toxinogenicity of *B. cereus* isolates surviving pasteurization that was reported to occur in 15% in our previous paper [15]. Comparison of predictions using different *B. cereus* toxinogenic strains might be a subject of the future research.

Since a high proportion of PBM is used in neonatal intensive care units (Table 3 and Table 4), we recommend using the second approach to risk assessment based on an estimation of the total number of CFU ingested per day per newborn.

## 5. Conclusions

Our method of risk assessment using our predictive model confirmed the safety PBM used in compliance with our SOP as well as of PBM compliant with the existing HMB microbiological safety post-pasteurization standards.

The predictions showed that thawed PBM should be used within 1 h of warming. Exceeding this time limit can lead to dangerous CFU concentrations.

For assessing the safety of PBM in neonatal intensive care units, we recommend using the metric of total *B. cereus* CFU ingested per day per newborn.

## Figures and Tables

**Figure 1 foods-11-01037-f001:**
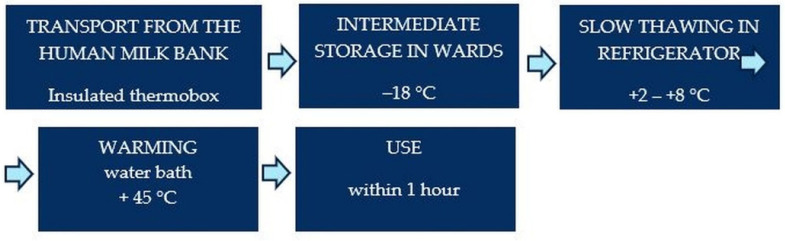
Schematic description PBM use after delivery.

**Figure 2 foods-11-01037-f002:**
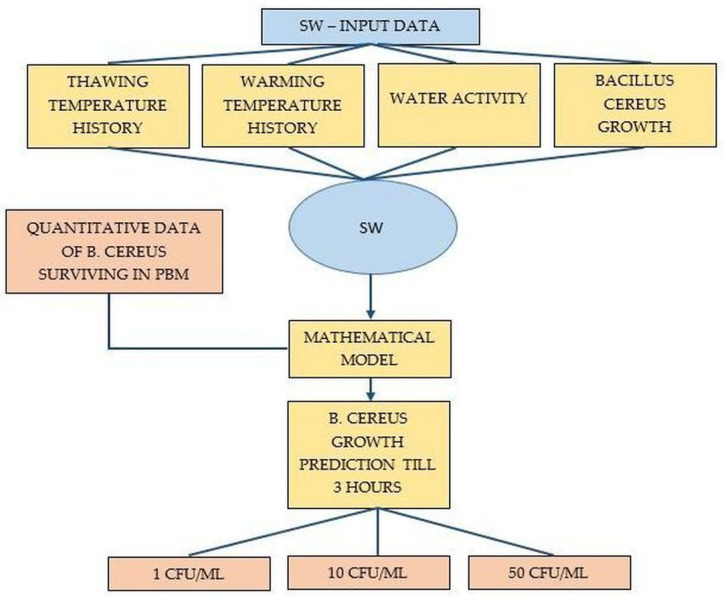
Schematic design of the experiment.

**Figure 3 foods-11-01037-f003:**
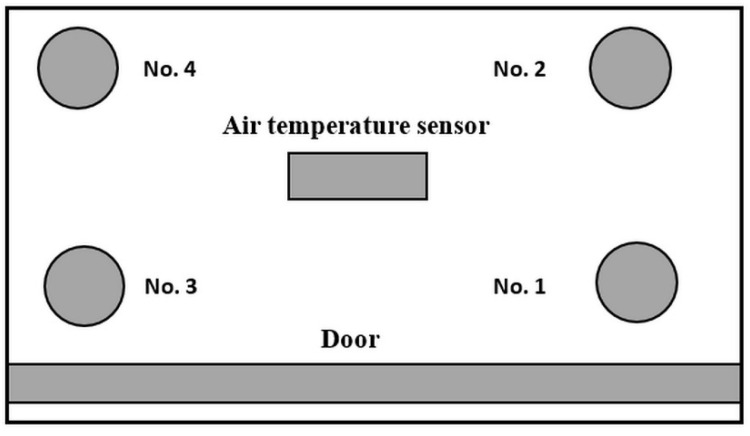
Position of 100 mL PBM bottles (No. 1–4) in the refrigerator during the measurement of the slow thawing temperature history, viewed from above.

**Figure 4 foods-11-01037-f004:**
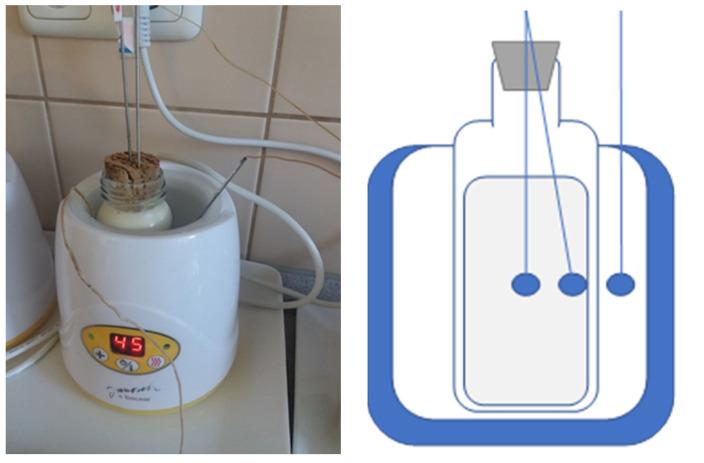
Temperature measurement during warming of bottles containing thawed PBM in a warming device adjusted to 45 °C (**left**); schematic of sensor positions (**right**).

**Figure 5 foods-11-01037-f005:**
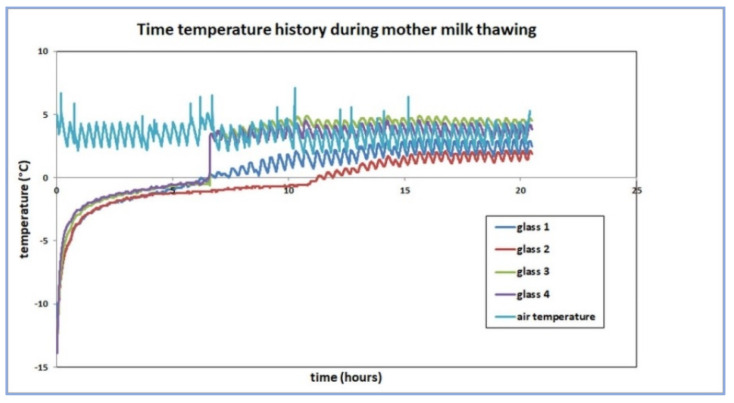
The temperature history of slow thawing of four 100 mL frozen PBM bottles in a refrigerator.

**Figure 6 foods-11-01037-f006:**
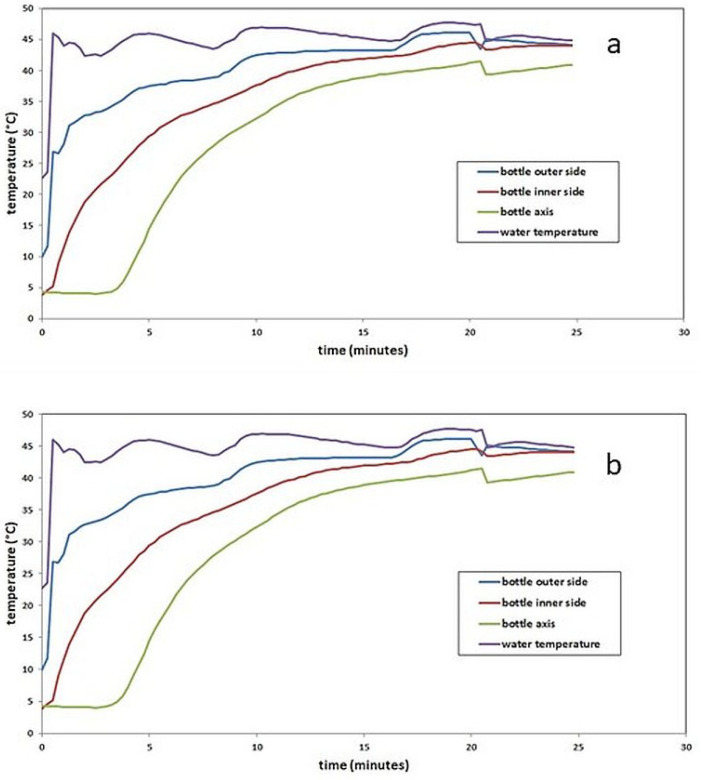
The temperature history of thawed PBM during warming of bottle No. 2 (**a**) and No. 4 (**b**).

**Figure 7 foods-11-01037-f007:**
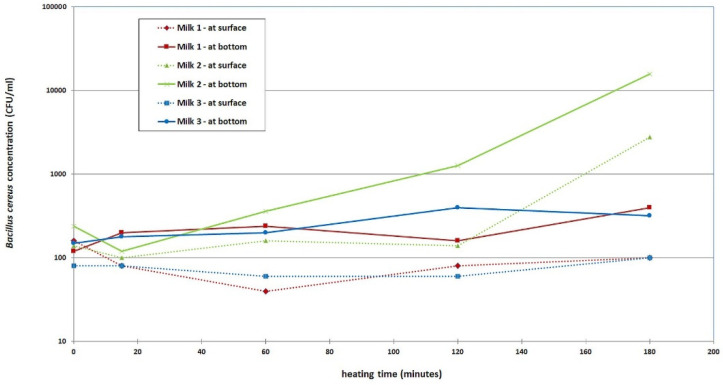
*B. cereus* growth during the warming of inoculated samples of thawed PBM.

**Table 2 foods-11-01037-t002:** Changes of *B. cereus* counts in thawed PBM during warming.

Bottle Code	Time of Warming (Minutes)	*B. cereus* Count (CFU/mL)
MM7595A	0	15
	15	15
	60	9
	120	8
	180	8
MM7599A	0	<1
	15	5
	60	13
	120	11
	180	25

**Table 3 foods-11-01037-t003:** PBM in 100 mL distribution bottles delivered from the Human Milk Bank to Pediatric departments.

Year	NICU	Neonatal Intermediate Care Unit	Healthy Newborn Unit	ICU	Bottles Total	Liters Total
**2017**	5196	8864	35	173	14,268	1426.8
**2018**	3839	8168	124	0	12,131	1213.1
**2019**	5166	8812	0	0	13,978	1397.8
**2020**	5491	12,069	0	0	175,860	1756.0
**Total**	19,692	37,913	159	173	57,937	5793.7
**Total (%)**	33.99	65.44	0.27	0.37	100.0	N.A.

**Table 4 foods-11-01037-t004:** PBM in 50 mL distribution bottles delivered from the Human Milk Bank to Pediatric departments.

Year	NICU	Neonatal Intermediate Care Unit	Healthy Newborn Unit	ICU	Bottles Total	Liters Total
**2017**	279	149	0	0	428	21.4
**2018**	897	451	56	0	1404	70.2
**2019**	970	332	0	0	1302	65.1
**2020**	683	13	0	0	696	34.8
**Total**	2829	945	56	0	3830	191.5
**Total (%)**	73.86	24.67	1.47	0	100.0	N.A.

**Table 5 foods-11-01037-t005:** The prediction of *B. cereus* CFU/mL concentration for an initial post-pasteurization concentration equal to the limit of detection and after 1, 2, and 3 h of storage at +42 °C (calculated mean values and the lower and upper limits of uncertainty).

InitialCFU/mL Count	Predicted Count (CFU/mL) after		
	1 h	2 h	3 h
1	1.5	6.0	69.0
**CFU/mL lower** **uncertainty limit**	1.2	1.7	3.7
**CFU/mL upper** **uncertainty limit**	3.6	2.6 × 10^2^	1.5 × 10^4^

Remark: *B. cereus* concentration values representing risk are labeled red.

**Table 6 foods-11-01037-t006:** The predicted number of *B. cereus* CFU ingested per day for initial post-pasteurization concentration of 1 CFU/mL after 1, 2 and 3 hours of warming.

Time of Warming of Thawed PBM (Hours)	Total PBM Volume Ingested per Day (mL)	Predicted CFU Number Ingested per Day	Lower CFU Number Uncertainty Limit	Upper CFU Number Uncertainty Limit
1	80	120	96	288
2	80	480	136	22,400
3	80	5500	296	12 × 10^6^

Remark: CFU number values representing risk are labeled red.

**Table 7 foods-11-01037-t007:** Prediction of *B. cereus* concentrations after 1, 2, and 3 h of storage at +42 °C for initial post-pasteurization concentrations of 10, 50, and 100 CFU/mL. Calculated values and the lower and upper limits of uncertainty are shown.

Initial CFU/mL Count	10	50	100
**1 h Predicted CFU/mL count**	15	75	150
**1 h CFU/mL lower uncertainty limit**	12	59	120
**1 h CFU/mL upper uncertainty limit**	36	180	360
**2 h Predicted CFU/mL count**	60.0	3.0 × 10^2^	6.0 × 10^2^
**2 h CFU/mL lower uncertainty limit**	17.0	87.0	1.7 × 10^2^
**2 h CFU/mL upper uncertainty limit**	2.6 × 10^3^	1.3 × 10^4^	2.6 × 10^4^
**3 h Predicted CFU/mL count**	6.9 × 10^2^	3.5 × 10^3^	6.9 × 10^3^
**3 h CFU/mL lower uncertainty limit**	37.0	1.9 × 10^2^	3.7 × 10^2^
**3 h CFU/mL upper uncertainty limit**	1.5 × 10^5^	7.9 × 10^5^	1.5 × 10^6^

Remark: *B. cereus* concentration values representing risk are labeled red.

**Table 8 foods-11-01037-t008:** The predicted number of *B. cereus* CFU ingested per day for initial post-pasteurization concentrations of 10 and 50 CFU/mL.

Initial CFU/mL Count	Total PBM Volume Ingested per Day (mL)	Total CFU Ingested per Day	Total CFU Lower Uncertainty Limit	Total CFU Upper Uncertainty Limit
10	80	1200	960	2880
50	80	6000	4720	14,400

**Table 9 foods-11-01037-t009:** Quantitative assessment of the post-pasteurization *B. cereus* contamination: compliance rate with different microbiological safety criteria.

Criterionlimit (CFU/mL)	Compliance Rate (%)	Source
Below 1	20	Own SOP, HMB standards of France, Australia, and USA [13]
Till 10	80	HMB standard of Italy, Sweden, and UK (Weaver 2019) [13]
Till 50	90	The norm for food used in children of age below 6 months [23]
Till 100	100	Codex Alimentarius (Cormontagne, 2021) [22]

## Data Availability

No new data were created or analyzed in this study. Data sharing is not applicable to this article.

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
