# Peer review of "Quantitative Risk Assessment of Bacillus cereus Growth during the Warming of Thawed Pasteurized Human Banked Milk Using a Predictive Mathematical Model"

_foods, 2022, doi:10.3390/foods11071037_

Round 1

Reviewer 1 Report

This is an interesting field of study. However, authors should make improvements in the Result Section

-Authors should describe what is observed in Figures and Tables in the Result Section.

-Figs 5, 6 and 7 can be merged into one figure containing a, b, c letters.

-Tables 5 and 6 can be redesigned and merged into one Table,

-Tables 7 and 11 can be redesigned and merged into one Table

-Tables 8, 9 and 10 can be redesigned and merged into one Table

Other comments:

-Line 71: What are the authors comparing to? The approach they used in the current study? If so, this has not been introduced yet and should be rephrase to clarify this paragraph.

-Line 83-85: Parragraph seemed confusing, as it sounded authors were referring to their previous work. In addition, they should better highlight the objectives of their study.

-Line 167: …was used for warming (Figure 4).

-Line 185: Do the authors refer to an ultra freezer as freezing box?

-Line 189: reference for Eph?

-Line 295-208: Authors should emphasize what methodology the software is based on their predictions.

-Table 1: No need to add “(-)” to water activity and pH. Why the pH of one of the samples is higher in comparison with others? (pH 7.3 vs 6.6). IS that a sign of spoilage?

Line 266: Authors should be consistent in what they are describing in text (hours) vs Figures (minutes).

Line 276: At what extent there was growth of B. cereus?

Line 288-292: Authors should describe what Tables 3 and 4 showed and should place these as supplementary tables instead.

Author Response

Reply to Reviewer 1

Dear reviewer,

we thank You very much for Your valuable comments and recommendations for improvements of our paper.

We have made revision of our manuscript according to Your advice.

-Authors should describe what is observed in Figures and Tables in the Result Section.

Reply: The result section was enlarged and several sentences were added in the section 3.7.

-Figs 5, 6 and 7 can be merged into one figure containing a, b, c letters.

Reply: The Figures 6 and 7 were put together (Figure 6, now). The Figure 5 remained, it describes the temperature history of slow thawing, Figures 6 and 7 describe the temperature history of thawed PBM during warming.

-Tables 5 and 6 can be redesigned and merged into one Table,

Reply: Tables 5 and 6 were put together as recommended (Table 5, now).

-Tables 7 and 11 can be redesigned and merged into one Table.

Reply: We decided to let the Tables 7 and 11 separated as Table 7 presents the detailed prediction of total CFU ingested per day in case of using different warming times while the Table 11 shows the results for 1-hour warming time only.

-Tables 8, 9 and 10 can be redesigned and merged into one Table

Reply: Tables 8, 9 and 10 were redesignated and merged into 1 table (Table 7, now).

Other comments:

-Line 71: What are the authors comparing to? The approach they used in the current study? If so, this has not been introduced yet and should be rephrase to clarify this paragraph.

Reply: Detailed explanation was added.

-Line 83-85: Parragraph seemed confusing, as it sounded authors were referring to their previous work. In addition, they should better highlight the objectives of their study.

Reply: We reformulated the sentence to clarify what was the previous work and what is the objective of the current work.

-Line 167: …was used for warming (Figure 4).

Done.

-Line 185: Do the authors refer to an ultra freezer as freezing box?

Reply: Yes, we do.

-Line 189: reference for Eph?

Reply: Reference was added.

-Line 295-208: Authors should emphasize what methodology the software is based on their predictions.

Reply: We reformulated the methods section concerning the Predictive Mathematical Model.

-Table 1: No need to add “(-)” to water activity and pH. Why the pH of one of the samples is higher in comparison with others? (pH 7.3 vs 6.6). IS that a sign of spoilage?

Reply: According to our experience there is a great variability of acidity of human milk (Jílková et al., 1980). For this reason we do not regard values of pH as sign of spoilage.

Jílková, V., Vávra, L., Měřička, P. (1981) Diagnostic Significance of Acidity Determination in Expressed Human Milk. In Les Banques de Lait Humain/Human Milk Banking Science et Technique du Froid  Refrigeration Science and Technology 1981-2, International Institute of Refrigeration, Paris, France p. 55-59 (ISBN 0151-1637).

Line 266: Authors should be consistent in what they are describing in text (hours) vs Figures (minutes).

Done.

Line 276: At what extent there was growth of B. cereus?

Reply: The achieved quantity of B. cereus was added.

Line 288-292: Authors should describe what Tables 3 and 4 showed and should place these as supplementary tables instead.

Reply: The Short explanation of the significances of the Tables 3 and 4 was added. Both tables were let in results section.

Reviewer 2 Report

The paper under consideration addresses the the risk evaluation of B. cereus growth during the processing of human milk from a breast milk bank through a predictive mathematical model.

It is based on a well-planned and detailed structured study, well described in the material and methods section, considering important factors such as post-pasteurization thawing and heating to the temperature of administration.

The results are well presented and discussed, presenting relevant conclusions for the practice of breast milk banks management and milk administration to newborns, namely for neonatal intensive care units.

So, I have no major issues or corrections to consider.

Author Response

Dear reviewer,

we thank You very much for Your encouraging comments.
